# Dynamic Changes of the Microbial Community and Volatile Organic Compounds of the Northern Pike (*Esox lucius*) during Storage

**DOI:** 10.3390/foods12132479

**Published:** 2023-06-25

**Authors:** Xuejiao Shang, Yabo Wei, Xin Guo, Yongdong Lei, Xiaorong Deng, Jian Zhang

**Affiliations:** 1School of Food Science and Technology, Shihezi University, Shihezi 832000, China; shangxuejiao-dy@163.com (X.S.); 18935813163@163.com (Y.W.); guoxin24yjs@163.com (X.G.); leiyongdong1008@126.com (Y.L.); dxr20099@163.com (X.D.); 2Key Laboratory for Processing and Quality Safety Control of Specialty Agricultural Products of Ministry of Agriculture and Rural Affairs (Provincial and Ministerial Cooperation), School of Food Science and Technology Shihezi University, Shihezi 832003, China; 3Key Laboratory for Food Nutrition and Safety Control of Xinjiang Production and Construction Corps, Shihezi University, Shihezi 832003, China

**Keywords:** spoilage, microbiota, *Esox lucius*, correlation, VOCs

## Abstract

In this study, the quality (sensory evaluation, microbial enumerate, color, tvb-n (total volatile basic nitrogen), tca-soluble peptide (trichloroacetic acid-soluble peptide), muscle glucose, lactate, total sugar, Bas (Biogenic amines), VOCs (volatile organic compounds) and the microbial dynamic structure in samples stored at 4 °C were evaluated, and the relationship between VOCs and the diversity structure of microorganisms was also discussed. It was determined by sensory evaluation that the shelf life of samples was around 8 days. Protein and sugar were detected in large quantities by microorganisms in the later stage. At the same time, this also caused a large amount of Bas (biogenic amines) (tyramine, cadaverine, and putrescine). According to high-throughput amplicon sequencing, the initial microbiota of samples was mainly composed of *Pseudomonas*, *Acinetobacter*, *Planifilum*, *Vagococcus*, *Hafnia*, *Mycobacterium*, *Thauera*, and *Yersinia*. Among them, *Pseudomonas* was the most advantageous taxon of samples at the end of the shelf life. The minor fraction of the microbial consortium consisting of *Vagococcus*, *Acinetobacter* and *Myroides* was detected. The substances 3-methyl-1-butanol, ethyl acetate, and acetone were the main volatile components. The glucose, lactic acid, and total sugar were negatively correlated with *Yersinia*, *Hafnia-Obesumbacterium*, *Thauera*, *Mycobacterium,* and *Planifilum*; the proportion of these microorganisms was relatively high in the early stage. TVB-N and TCA-soluble peptides were positively correlated with *Pseudomonas*, *Shewanella*, *Brochothrix*, *Vagococcus*, *Myroides,* and *Acinetobacter*, and these microorganisms increased greatly in the later stage. The substance 3-methyl-1-butanol was positively correlated with *Pseudomonas* and negatively correlated with *Mycobacterium*. Ethyl acetate was associated with *Hafnia-Obesumbacterium*, *Thauera,* and *Yersinia*. Acetone was positively correlated with *Acinetobacter*.

## 1. Introduction

The northern pike (*Esox lucius*), as the top predatory species of the fish community, represents the most socio-economic freshwater fish species. They can be found in the cold and temperate regions of the northern hemisphere [1]. The northern pike is popular with consumers because of its fast growth rate, high nutritional value, and excellent taste. However, the fish is highly susceptible to spoilage, as it is high in water content, rich in nutrients, and neutral in pH value and has high endogenous enzyme activity [2]. Previous studies have shown that the spoilage of fish is related to their microbiota composition [3,4,5,6,7]. Nevertheless, microbiota in fish alters dramatically with storage time, species, environment, processing operation and other factors, so it is different at different storage stages [3,5,8,9,10]. The initial microbiota of fish consists of microorganisms from water, capture, treatment, equipment, workers and others. Previous studies have shown that the species composition of fresh grass carp [11], cod loin [12] and raw Atlantic cod [5] included *Acinetobacter*, *Aeromonas*, *Brevundimonas*, *Chryseobacterium*, *Cutibacterium*, *Empedobacter*, *Flavobacterium*, *Geobacter*, *Glutamicibacter*, *Janthinobacterium*, *Lactococcus*, *Photobacterium*, *Pseudomonas*, *Psychrobacter*, *Microbacterium*, *Moraxella*, *Shewanella* and *Soonwooa*. However, only some microorganisms are able to adapt to harsh environments, grow and reproduce. *Acinetobacter*, *Aeromonas*, *Chryseobacterium*, *Flavobacteriaceae*, *Micrococcus*, *Macrococcus*, *Photobacterium*, *Pseudomona*, *Psychrobacte* and *Shewanella* are the most common freshwater fish microorganisms [3,13,14,15,16] and can secrete some enzymes to soften fish, produce small molecule peptides [17], free amino acids, biogenic amines and other metabolites [18] and finally produce volatile organic compounds (VOCs) such as hydrocarbons, esters, aldehydes, ketones, acids and others [19], resulting in a decrease in the sensory quality of fish. The dominant microorganisms in the process of aquatic product spoilage are called SSOs (specific spoilage organisms). The number and community proportion of SSOs in the early storage period of aquatic products are very small, but with the extension of storage time, the number of SSOs increases rapidly, and the proportion of colonies increases obviously [20,21]. Considering that, it is necessary to study the relationship between spoilage organic compounds and volatile organic compounds to identify spoilage markers and to control fish spoilage.

There are no studies to research *Esox lucius* microbiota using an HTS (high-throughput amplicon sequencing) approach and potential spoilage markers. Therefore, in this study, the diversity of the microbial community in samples during storage at 4 °C was evaluated by using HTS technology, the quality changes and VOCs of the samples were determined, and the correlation between microorganisms and volatile organic compounds was further analyzed in order to provide a reference for the preservation of fish fillets under cold storage conditions.

## 2. Materials and Methods

### 2.1. Samples Preparation

*Esox lucius* (the number is 15; approx. 1000 g) were purchased from an aquatic products market in Urumqi, Xinjiang, China (87.553646° N, 43.819834° E), and then transported to the laboratory alive. The fish were stunned, scaled, gutted, skinned, filleted, and washed with cold sterile water immediately. The fish was cut into pieces of the same size and thickness (about 2 × 3 × 1.5 cm^3^). Each piece was about 10 ± 2 g and packaged individually in a sterile polyethylene bag. Then fish filets were stored in a refrigerator at 4 ± 0.5 °C. Sensory evaluation and total viable counts (TVC) were measured at days 0, 1, 3, 5, 7, 9, 11 and 13. VOCs and microbial community analysis were carried out at days 1, 5, 9 and 13.

### 2.2. Sensory Analysis

The sensory analysis of samples referred to the method of [22]. Briefly, the sensory acceptability test of samples was evaluated by a trained group of three male and three female laboratory members aged 21 and 35, according to a five-point scale. A score of 4.0–5.0 indicated good quality, 3.0–4.0 indicated acceptable quality, 2.0–3.0 indicated unacceptable quality, and 0.0–2.0 indicated an intense dislike and completely spoiled state. The score of 3.0 was the division between sensory acceptance and sensory unacceptance.

### 2.3. Microbial Enumeration

The total number of spoilage microorganisms in the samples was counted by the plate counting method. Each 10 g sample was weighted, homogenized and mixed with 90 mL (1:10, *v*/*v*) of sterile physiological saline (0.85%, *w*/*w*). Then, the homogenate was further serially decimally diluted, and the appropriate gradient was selected to count the total viable counts (TVC) by plate count agar (containing 0.5% tryptone, 0.25% yeast extract powder, 0.1% glucose and 1.5% agar) at 25 ± 1 °C for 48–72 h. Plates with 30 to 300 bacterial colonies were used to determine the TVC of the sample. The results were expressed as log CFU/g.

### 2.4. Color

The values of L*, a*, and b* were measured by a handheld colorimeter (Model No. SR-60, Shenzhen Threenh Technology Co., Ltd. Shenzhen, China) (detection method reference manual). The L* value indicates lightness, a* and b* are chromaticity coordinates. On the color space diagram, L* is represented on a vertical axis with values from 0 (black) to 100 (white). The a* value indicates the red (+)-green (−) component of a color, the b* value indicates the yellow (+)-blue (−) components of an instrument operation reference manual.

### 2.5. TCA-Soluble Peptide Contents

We used the extraction and determination method of TCA-soluble peptides in samples described by Zhuang and co-authors [22] with appropriate adjustments. The 3 g sample was homogenized with 27 mL 5% (*w*/*v*) cold trichloroacetic acid, then the homogenized solution was placed at 4 °C for 30 min and centrifuged at 8000× *g* for 10 min. The supernatant was diluted and determined by Lowry’s method and expressed as μmol tyrosine/g muscle.

### 2.6. Total Volatile Basic Nitrogen (TVB-N) Contents

The concentration of total volatile basic nitrogen (TVB-N) was determined by the current Chinese national food safety standard (GB 5009.228-2016). A 20 g sample was mixed with 100 mL water and shaken for 30 min, and the mixed solution was allowed to stand at 4 °C for 30 min and filtered. A 5 mL 10 g/L magnesium oxide solution was added to 10 mL filtrate, and the content of TVB-N was determined by the semi-micro-kjeldahl method. A 10 mL 20 g/L boric acid solution was added into the receiving container to adsorb the distillate. Titration of the boric acid solution was with 0.01 M HCl, and the TVB-N content was calculated according to Equation (1). At the same time, do the reagent blank.
(1)$$X=\frac{(V1-V2)\times c\times 14}{m\times (V/V0)}$$
where X: the content of the volatile basic nitrogen in the sample, mg/100 g;V1: the volume of the standard titration solution of hydrochloric acid consumed in the sample, mL;V2: the reagent blank that consumes the volume of the standard titration solution of hydrochloric acid, mL;C: the concentration of the standard titration solution of hydrochloric acid, mol/L;14: titrate the equivalent nitrogen mass of 1.0 mL hydrochloric acid c(HCl) = 1.000 mol/L standard titration solution, g/mol;M: the mass of the sample, g;V: the accurately sucked filtrate volume, mL; andV0: the total volume, mL.

### 2.7. Muscle Glucose, Lactate and Total Sugar Contents

The muscle glucose content was determined using the glucose oxidase method, following the instructions of a glucose assay kit (Shanghai Rongsheng Biotech Co., Ltd. Shanghai, China). Briefly, 1.0 g of fish flesh homogenized with 2.0 mL deionized water was bathed in boiled water for 5 min so as to inactivate endogenous enzymes related to glucose metabolism. After that, the homogenate was centrifuged at 2000× *g* for 3 min. The resulting supernatant was used for muscle glucose determination, and the final results were expressed as μmol glucose/g muscle. 

According to the instruction of the corresponding test kit (No. A019-2-1, Nanjing Jiancheng Bioengineering Institute, Nanjing, China), determination of lactate concentration was conducted using 10% muscle tissue homogenates (in 0.85% physiological saline) by the lactate dehydrogenase method. Concentration of muscle lactate was expressed as μmol lactate/g muscle. 

The total sugar content in fish flesh was determined under the guidance of a commercially available test kit (No. BC2715, Beijing Solarbio Science & Technology Co., Ltd. Beijing, China). The determination process was performed using the DNS (3,5-dinitrosalicylic acid) regent colorimetric method, which involved the acidic hydrolysis of polysaccharides in fish flesh into reducing sugar, the reaction of reducing sugar with the DNS regent, and finally the absorbance measurement at 540 nm. The result was expressed as mg glucose (equivalent)/g muscle.

### 2.8. Biogenic Amines Contents

We used the extraction method of biogenic amine in samples described by Zhuang and co-authors [22] with appropriate adjustments. A 5 g sample was placed in a centrifuge tube, followed by 10.0 mL 0.4 M cold perchloric acid (PCA) solution. The above two were homogenized and centrifuged at 10,000× *g* for 10 min. After collecting the supernatant, the precipitation was washed with 10.0 mL 0.4 M cold PCA solution and then centrifuged at 10,000× *g* again. All supernatants obtained from the two centrifugations were combined and added to 25 mL with 0.4 M PCA solution. The derivative steps and further analysis of biological amines referred to the method of Zhuang et al. [22]. This involved a precolumn derivatization with a chromatographic analysis using HPLC (Agilent Technologies Co., Ltd., Santa Clara, CA, USA) equipped with an Inertsil C18 column (4.6 mm × 250 mm × 5 µm) and an ultraviolet detector set at 254 nm. Biogenic amines were eluted using water (solvent A) and acetonitrile (solvent B) as mobile phases. The gradient elution was 0 min, 10% B; 5 min, 65% B; 20 min, 100% B; 24 min, 100% B; and 40 min, 65% B. The sample was eluted at 30 °C with an injection volume of 20 μL. Biogenic amines were determined and quantified based on the retention time and peak area compared with standard solutions.

### 2.9. Volatile Organic Compounds (VOCs)

The VOCs of samples were analyzed using a solid-phase microextraction system (57330-U, Supelco, Inc., Bellefonte, PA, USA) containing fiber coated with a 50/30 μm thickness of DVB/CAR/PDMS (divinylbenzene/carboxen/polydimethylsiloxane) combined with an Agilent 7890A gas chromatograph/an MS-5975C mass spectrometer equipped with an HP-5MS column (60 m × 250 μm, film thickness, 0.25 μm) according to the method described by [23,24]. The extraction needle was activated for 20 min (250 °C) at the gas inlet before use. A total of 5 g minced sample and 100 μL mixed working internal standard solution were placed in a 20 mL headspace vial and enclosed immediately. Then, the VOCs were released at 85 °C for 20 min and extracted at isothermal temperature for 30 min with an extraction needle. The VOCs were separated using a GC/MS system. The GC conditions were as follows: the initial temperature was maintained at 50 °C for 1 min, raised to 100 °C at a rate of 5 °C/min and maintained at 100 °C for 2 min, raised at 180 °C at a rate of 4 °C/min and maintained 180 °C for 3 min, and raised at 250 °C at a rate of 5 °C/min and maintained 250 °C for 5 min. The inlet temperature was 250 °C, and the interface temperature of GC was 280 °C with the split ratio of 4:1 and the flow rate of 1.5 mL/min (carrier gas: helium). The working conditions of the MS were set as follows: ionization energy at 70 eV; scan range at 35–550 *m*/*z*; and ion source temperature at 230 °C. The results were qualitatively analyzed using NIST 17.0 (National Institute of Standards and Technology, Gaithersburg, MD, USA). Finally, the relative content of the VOCs that were identified was calculated using area normalization.

### 2.10. The Analysis of Microbial Composition

The following procedures were completed: Mix 5 g sample and 50 mL sterile saline, homogenized for 5 min. Filter, take the supernatant, centrifuge at 24,200× *g* for 10 min, discard the supernatant. The sediment was dissolved with sterilized water, and the total bacterial DNA was extracted using the Bacterial Genome DNA Miniprep Kit (Axigen Scientific Inc., Union City, CA, USA) to extract total bacterial DNA. The quantification and purity of DNA were evaluated by an ultra-micro ultraviolet spectrophotometer and agarose gel electrophoresis.

The DNA was used as the template, and the primers 338 F (5′-ACTCCTACGGGAGGCAGCAG-3′) and 806 R (5′-GGACTACHVGGGTWTCTAAT-3′) were used to amplify the V3-V4 hypervariable regions of 16S rRNA genes by the thermocycler PCR system according to the reported method. The PCR amplification system included the following: 20 μL: 5× fastPfu Buffer 4 μL, dNTPs (2.5 mM) 2 μL, forward primer (5 μM) 0.8 μL, reverse primer (5 μM) 0.8 μL, fastPfu Polymerase 0.4 μL, BAS 0.2 μL, DNA template 10 ng, add ddH_2_O to 20 μL. The amplification parameters were as follows: initial denaturation at 95 °C for three minutes, denaturation at 95 °C for 30 s, annealing at 55 °C for 30 s, extension at 72 °C for 45 s, and final extension at 72 °C for 10 min, for 29 cycles. The PCR products were recovered and purified by agarose gel. Finally, the high-throughput sequencing platform was used for sequencing. The bands passing the quality inspection were sequenced continuously on the Illumine MiSeq platform of Shanghai Majorbio Bio-pharm Technology Co., Ltd. (Shanghai, China). After the high-throughput amplicon sequencing, raw reads were processed according to [25]. First, quality control was carried out after sequence splicing. The biological information of the sequences qualified in quality control was analyzed by the QIIME (1.9.1) platform, and the aligned sequences were processed by Py NAST (http://pynast.sourceforge.net) software. According to the similarity of 100% and 97%, the two-step UCLUST method was used to construct the matrix of operational taxonomy units (OTUs). An OTU containing chimera was selected and removed by Chimera Slayer software, and then representative OTU sequences were selected and compared in the ribosomal ribonucleic acid (RNA) database RDP, SILVA and the Greengenes database to determine their different taxonomic status. Illuminane hiseq platform and http://www.bioinformatics.com.cn/ (accessed on 10 September 2022).

### 2.11. Statistical Analysis

Differences of means in physical and chemical indicators and microbial indicators were repeated at least three times. Experimental data were analyzed using one-way analysis of variance (ANOVA) in SPSS 26.0 (SPSS Inc., Chicago, IL, USA), and significance was determined by Duncan’s test and expressed using different lowercase letters. A significant difference was indicated as *p* < 0.05. To clarify the relationship between microorganisms and protein, carbohydrates and VOCs, correlation analysis was determined by the Pearson correlation test.

## 3. Results

### 3.1. Sensory Evaluation

The sensory acceptability score of samples decreased gradually during storage (Figure 1A). On day 0, all samples had high sensory scores, demonstrating excellent quality. After 9 days of storage, the fillets became sensory unacceptable. The fish fillets completely deteriorated on day 13, and the lowest sensory score was 2.63. According to the time of sensory rejection, we divided cold storage into two stages (the early stage from day 1 to day 7, and the late stage from day 9 to day 13).

### 3.2. Color Change

The change in the chromaticity value of fish was an important indicator of its freshness. As shown in Table 1, during the storage period, the value of L* of the sample increased, indicating a significant change in brightness, the value of a* showed a clear downward trend (*p* < 0.05) and the value of b* showed an upward trend. During the storage of fish, the growth and metabolism of microorganisms, the degradation of protein and the oxidation of fat have effects on the color of fish. In the process of growth and reproduction, microorganisms can secrete special substances to cause color changes in fish. Myoglobin in fresh fish remains in a reduced state, and myoglobin will gradually oxidize and affect the color of fish with the extension of storage time after slicing [26]. At the same time, the fat in fish will also be oxidized during storage, which will lead to darkening and yellowing.

### 3.3. Biochemical Change

The initial TCA-soluble peptide content in the sample was 1.56 μmol/g (Figure 1C). The TCA-soluble peptide concentration of samples increased continuously during storage. From day 1 to day 7, the level of samples was relatively stable, followed by a rapid increase, reaching 8.37 μmol/g on day 13. This indicated that the protein had undergone serious degradation and produced a large amount of soluble peptides after the 7th day.

The content of TVB-N is usually used to determine the freshness of fish. As is shown in Figure 1D, the level of TVB-N showed an obvious rising trend, especially after 7 days. The concentration of TVB-N exceeded the limit (30 mgN/100 g) on day 9.

Changes in the mussel glucose content of samples are shown in Figure 1E. The concentration of mussel glucose with the extension of storage time showed a trend of decline, especially from day 1 to day 7, with the declining rate of glucose content being faster at this stage. This indicated that glucose was the main substance used by microorganisms in prophase.

The lactic acid content changed little from day 1 to day 7, from 5.90 μmol/g to 5.08 μmol/g (Figure 1F). This may be due to the low utilization of lactic acid by microorganisms and the increase in lactic acid in glycolysis. Then, the concentration of lactic acid decreased rapidly to 3.19 μmol/g on day 13. It was shown that a large number of microorganisms produced in samples accelerated the utilization of lactic acid. Notably, the rate of decline of lactic acid increased as glucose was depleted, suggesting that microbes in samples might use lactic acid as a carbon source when glucose is depleted. Sofos et al. [27] also demonstrated that lactic acid could be an energy source for post-glucose metabolism during spoilage in meat and seafood. As shown in Figure 1G, the initial total of sugars in the sample was 1.69 mg/g, which dropped to 0.33 mg/g on day 7, possibly due to microbial depletion of glucose in the sample. Subsequently, the total sugar content rapidly decreased to a minimum of 1.08 mg/g on day 13, indicating that the bacteria had high activity utilizing carbohydrates during spoilage. The glucose content remained low and stable after day 7; the rapid decline in total sugars might be due to the utilization of other carbohydrates by microorganisms, such as ribose and glycogen.

### 3.4. BAs

Table 2 indicated the changes of BAs of samples during storage. The contents of BAs of fresh fish were low. In addition to tryptamine, BA content increased gradually with the extension of the storage time. 

Tyramine is a kind of BA with strong toxicity. In the early stage of storage, the content of tyramine was second only to cadaverine and putrescine. However, when the samples were stored on day 13, the content amounts reached 100.96, 85.44 and 73.23 mg/kg, respectively, which were the high content of BAs in fish. This showed that tyramine, cadaverine and putrescine were the main BAs produced in samples at 4 °C.

Spermidine and spermine are the natural ingredients of living cells [28]. The contents of spermidine and spermine in fresh fish were high, and they were the highest on day 7, then began to decrease slightly. The initial increase of spermidine and spermine might be related to free arginine decarboxylation caused by bacterial decarcase. However, the spermidine and spermine decreased on day 9—they reached 3.20 and 4.12 mg/kg, respectively. This might be because they were utilized by microorganisms as nitrogen sources [29].

### 3.5. Microbiome Analysis

#### 3.5.1. Total Viable Counts (TVC)

As is shown in Figure 1B, the TVC of the samples rapidly increased from the initial value of about 3.4 lgCFU/g to 8.15 lgCFU/g by the end of storage. After storage for 5 days, the TVC increased to 5.83 lgCFU/g, increasing by 71.47% compared with day 1. However, with the limit of 7.0 lgCFU/g for freshwater fish [30], the TVC of samples reached the limit (7.0 lgCFU/g) from 7 days to 9 days. After 11 days, the growth rate of microorganisms entered a stable period, an increase of 4.75 lgCFU/g compared with day 1. Therefore, the shelf life of aerobic refrigerated fillets may be 8 days in terms of microbial count. Days 1, 5, 9, 13 were selected for subsequent experimental analysis.

#### 3.5.2. Community Abundance and Diversity

High-throughput sequencing was used to analyze the composition of bacterial species in fish fillets during storage periods. A total of 194,821 reads were generated, with an average of 48,705 reads per sample. The number of bacterial readings ranged from 44,771 on sample 13d to 52,940 on sample 5d (Table 3). Based on the results of the OTUs, Venn graphs visualized the similarities and differences of microbial diversity among samples across the entire storage period (Figure 2). There were 13 core OTU groups in different storage stages, and 796 unique OTUs were shown on day 1 and sharply reduced to 0 on day 13. The results showed that the microbial diversity of the samples decreased with the extension of time during storage.

As shown in Table 3, the Good’s coverage value was more than 99%, indicating that the sample had a high coverage rate, reflecting that the sequencing results represented the real diversity of the microbes in the sample. The ACE and Chao1 indices are usually used to reflect the community richness of bacteria, and the higher the indices, the higher the community richness [31,32]. The Chao1 and ACE indices (Table 3) of bacteria gradually decreased during the entire storage process, indicating that the richness of microbial communities was lowest on day 13. The Simpson index was negatively correlated with other diversity indices. During the storage period, the Shannon index decreased gradually, and the Simpson index increased gradually. In addition, the higher Shannon index showed that the diversity of bacteria was lower [33,34]. Therefore, the species in samples decreased gradually with the extension of storage time, which may be related to the interaction between microorganisms during storage [35]. 

The rank-abundance curve can directly reflect the abundance and uniformity of species in the sample. The wider the curve, the higher the species richness of the sample; the smoother the curve, and the greater the uniformity of the sample(Figure 3). The curve width was the smallest and smoothest on days 9 and 13, which represented that the abundance and uniformity of the samples at the end of storage was the smallest. The results showed that the diversity of microorganisms decreased during the late storage period, and some dominant bacteria might be produced, which may be related to temperature, storage time and the nutritional composition of the fish flesh [8,36].

#### 3.5.3. Relative Abundance of the Major Phyla and Genera

The relative abundance of bacteria at phylum and genus levels was obtained using high-throughput sequencing analysis to understand the diversity and composition of microbial communities during storage (Figure 4). Samples consisted mainly of Proteobacteria, Firmicutes, Actinobacteriota and Bacteroidota at the phylum level (Figure 4A). The proportion of Proteobacteria changed most significantly during storage, especially on the 9th day, when the content reached 94.56%, while the other members were present in minor percentages. Figure 4B showed the relative abundance of bacteria in all samples at the genus level. The composition of the microbial community mainly included *Pseudomonas*, *Acinetobacter*, *Planifilum*, *Vagococcus*, *Shewanella*, *Hafnia*, *Brochothrix*, *Mycobacterium*, *Myroides*, *Thauera* and *Yersinia*. During storage of samples, *Pseudomonas* presented an obvious upward trend, represented on 0.55% (sample of 1d) and 75.86% (13d) of sequences, indicating that *Pseudomonas* was the predominant genera and played key roles in the quality of fillets. Some researchers have reported that *Pseudomonas* is the main spoilage microorganism of freshwater fish during cold storage, and it can produce a variety of undesirable volatile odors, making its sensory output unacceptable [37,38].

### 3.6. VOCs

In this study, VOCs in samples were detected on days 1, 5, 9 and 13 by GC-MS. A total of 25 VOCs were detected, which mainly included alcohols, aldehydes, ketones, acid, esters, sulphur compounds and other compounds (Appendix A). By comparing the composition and content of four groups of VOCs, we determined which compounds increased, decreased, disappeared or fluctuated during storage (Figure 5). 

Types and concentrations of VOCs changed at the later stage of storage compared with initial VOCs. A total of 5 alcohols were detected in experimental groups; ethanol was high in samples but usually had only a minor effect on food flavor [39]. The substance 3-methyl-1-butanol appeared in samples on days 9 and 13. The relative content reached 23.49% and 22.11% respectively. The substance 3-methyl-1-butanol produces an odor similar to that of whiskey- and is derived mainly from the catabolism of leucine or isoleucine. It has been recognized as a potential spoilage indicator of fish [40,41]. The degradation of phenylalanine mainly includes benzyl alcohol, phenylethyl alcohol, benzaldehyde, phenylacetaldehyde and other small volatile compounds [42]. Phenylethyl alcohol and benzyl alcohol increased progressively during storage. The level of enzaldehyde was higher than benzyl alcohol and phenylethyl alcohol, and the early level of enzaldehyde was higher than later ones.

Acids are mainly produced by hydrolysis of triglycerides and phospholipids [43]. Acetic acid was just detected on day 9 and 13. The level of n-hexadecanoic acid decreased during storage. Especially in the later stage, the content of n-hexadecanoic acid was significantly lower than acetic acid. The substances 2,3-butanedione and 2-pentanone were not produced in fresh fish, which indicates that two ketones are highly putrid volatile markers of fish. Acetone is produced by the catabolism of glucose [44]. With the extension of storage time, the content of this compound increased. Microorganisms might involve the utilization of glucose during storage. Esters are usually formed through esterification of alcohols and carboxylic acids, and ethanol seems to be used as a preferred substrate for ester synthesis [45]. Ethyl acetate amounts were high at each stage. This might be related to the high level of ethanol detected in this study. The sulfur-containing compounds is usually related to the utilization of sulfur-containing amino acids by microorganisms and plays in an important role in the flavor of fish [46].

### 3.7. Correlation of Microorganisms with Protein, Carbohydrates and VOCs 

During storage, microbial metabolism of carbon source compounds and nitrogen source compounds plays an important role in fish spoilage. Carbon source compounds mainly come from glycogen, glucose and lactic acid in muscle, while nitrogen source compounds mainly come from protein in fish. In particular, the increase of free amino acids, TVB-N and TCA- soluble peptides is related to the degradation of protein. Figure 6 shows the correlation of microorganisms with protein and carbohydrates during storage. The glucose, lactic acid and total sugar were negatively correlated with *Yersinia*, *Hafnia-Obesumbacterium*, *Thauera*, *Mycobacterium* and *Planifilum*; the proportion of these microorganisms were relatively high in the early stage, but they gradually decreased in the later stage. TVB-N and TCA-soluble peptides were positively correlated with *Pseudomonas*, *Shewanella*, *Brochothrix*, *Vagococcus*, *Myroides* and *Acinetobacter*, and these microorganisms increased greatly in the later stage. The results showed that carbohydrates were preferentially utilized by microorganisms in the early stage of storage, and protein was mainly consumed in the later period.

The correlation between VOCs and microorganisms is shown in Figure 7. *Pseudomonas*, *Acinetobacter*, *Hafnia*, *Thauera* and *Yersinia* were positively correlated with VOCs. *Planifilum* was negatively correlated with VOCs. *Pseudomonas* plays a major role in the production of alcohols [40]. It occupied the main dominant position in samples. *Pseudomonas* was positively correlated with phenylethyl alcohol, 3-methyl-1-butanol and acetic acid (*p* < 0.05). This was the opposite of *Mycobacterium*. *Mycobacterium* and Thauera were positively correlated with n-hexadecanoic acid and methylene chloride. *Acinetobacter* showed a significant positive correlation with acetone (*p* < 0.05). *Planifilum* was significantly associated with 1-butanol (*p* < 0.05). *Hafnia* and *Yersinia* were positively correlated with toluene (*p* < 0.01), carbon disulfide, phenol, p-xylene, ethyl acetate and styrene (*p* < 0.05). 

## 4. Discussion

Through sensory evaluation, color, tvb-n, tca-soluble peptide, muscle glucose, lactate, total sugar, BAs, VOCs and microbial analysis, including the combination of high-throughput sequencing and SPME-GC-MS technology, the quality, bacterial community and composition of VOCs were explored, and the microbial spoilage of samples stored at 4 °C was studied. The correlation between microbiome and protein, carbohydrates and VOCs and its relationship with the sensory acceptability score enable us to understand how the quality of fish deteriorates during storage and how it is eventually rejected.

This study clarified that the sensory acceptability of fish was unacceptable at day 8. The fish turnedj green and yellow. Protein degraded drastically after the 7th day. Specifically, there was a sharp increase in the content of TVB-N and TCA-soluble peptides. The main use of microorganisms in fish in the early storage period may be glucose, and in the later period, it is mainly lactic acid. This is because the total sugar of samples decreased with the increase of storage time. In particular, the glucose content decreased rapidly in the early stage and then slowed down. However, the change of lactic acid is just the opposite. A total of 8 types of BAs can be detected in fresh fish. The content of spermidine and spermine are relatively high. When the fish has deteriorated on the 8th day, tyramine, cadaverine and putrescine become the main BAs.

The study studied diversity and succession of microbial communities during fish storage. The main bacteria produced during storage were Proteobacteria and Firmicutes, and the main bacteria were *Pseudomonas*, which was predictable. Previous studies have shown that *Pseudomonas* and H_2_S producing bacteria are usually the cause of the quality loss of aquatic products [47,48,49,50]. *Pseudomonas* is the main spoilage microorganism of aquatic products under aerobic conditions [51] and has a strong ability to decompose protein and fat, resulting in the release of bad smell from fish, which is ultimately unacceptable to the senses [52]. In this study, at the end of storage at 4 °C, in addition to *Pseudomonas*, the microbiota present also included *Vagococcus*, *Acinetobacter*, *Brochothrix*, *Myroides* and *Shewanella*, and the microorganisms such as *Vagococcus*, *Acinetobacter* and *Myroides* detected at the beginning of storage were usually related to pretreatment. These pollution sources may come from workers, processing equipment, air, water, etc. [53,54,55]. Other bacteria from human and environmental sources, such as *Enterobacter*, *Enterococcus*, *Escherichia coli*, *Klebsiella*, *Shigella* and *Staphylococcus*, were also commonly produced during capturing, processing and freezing, affecting the quality and safety of fish. The relative abundance of these genera was low in this paper, but that does not minimize the possibility of the existence of pathogens. They all contain human pathogens, which may be found in raw fish whether they are foodborne or not. In this case, eating undercooked fish may lead to food-borne infections. *Brochothrix* and *Shewanella* were not detected at the initial stage, but the relative content reached 5.57% and 0.42% at the end. *Brochothrix* is a common spoilage bacterium in meat [56]. *Shewanella* is a gram-negative psychrophilic bacterium, which can produce H_2_S, has the ability to reduce oxidized trimethylamine to trimethylamine and produce a variety of biogenic amines, and is the main potential spoilage microorganism of protein-rich aquatic products [57,58]. It was found that during storage, *Planifilum*, *Mycobacterium*, *Thauera* and many unknown bacteria decreased and were gradually replaced by *Pseudomonas*.

In our study, the content of ethanol and ethyl acetate was high. Some studies have proven that there is a certain relationship between ethanol and the formation of esters, and 3-methyl-1-butanol is often observed during the deterioration of seafood and is a suitable deterioration index for fish [38,59,60]. The substance 3-methyl-1-butanol was detected on days 9 and 13. The 3-methyl-1-butanol produced from isoleucine is the important peculiar smell of spoiled cold smoked salmon [61,62]. *Pseudomonas* is related to the synthesis of 3-methyl-1-butanol, trimethylamine (TMA) and esters [38,63]. *Brochothrix thermophacta* may be involved in the synthesis of acetylcarnosine; *Shewanella* can reduce the trimethylamine oxide (TMAO) present in fish to TMA [64] and was not observed in this study.

## 5. Conclusions

This study clarified the change of quality and the diversity and dynamic changes of the microbial community of samples during cold storage and analyzed the volatile components produced during storage and the correlation between them. During storage, the quality of fish was decreased, the sensory score declined, fish films turned green and yellow, and there was also protein degradation, sugar content decrease and tyramine, cadaverine and putrescine accumulation. The most abundant microbial taxa were Proteobacteria, Firmicutes and Bacteroidea at the phylum level, and *Pseudomonas* was the dominant genus. Alcohols and esters were the main VOCs in fish of spoilage. Ethanol, 3-methyl-1-butanol, ethyl acetate, methylene chloride, acetone, toluene, ethylbenzene and carbon disulfide were identified and quantified as the potential chemical corruption index of samples stored aerobically using statistical multivariable methods. Phenylethyl alcohol, 3-methyl-1-butanol and 1-butanol were associated with *Pseudomonas*, *Planifilum* and *Mycobacterium*. Ethyl acetate was associated with *Hafnia*, *Thauera* and *Yersinia*. *Pseudomonas* is suggested as the main SSO to help predict the shelf life of critical levels and may be related to 3-methyl-1-butanol.

## Figures and Tables

**Figure 1 foods-12-02479-f001:**
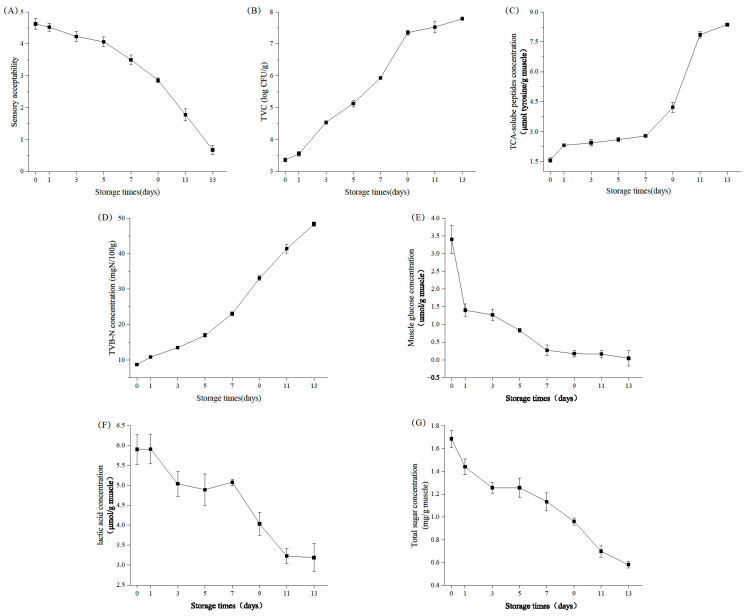
Changes in sensory evaluation scores (**A**), total viable count (**B**), TCA-soluble peptides concentration (**C**), TVB-N concentration (**D**), mussel glucose concentration (**E**), lactic acid concentration (**F**), total sugar concentration (**G**) of flesh during cold storage (sensory acceptability scores: 4.0–5.0 = good quality, 3.0–4.0 = acceptable quality, 2.0–3.0 = unacceptable quality, 0.0–2.0 = completely spoiled and strongly disgusting).

**Figure 2 foods-12-02479-f002:**
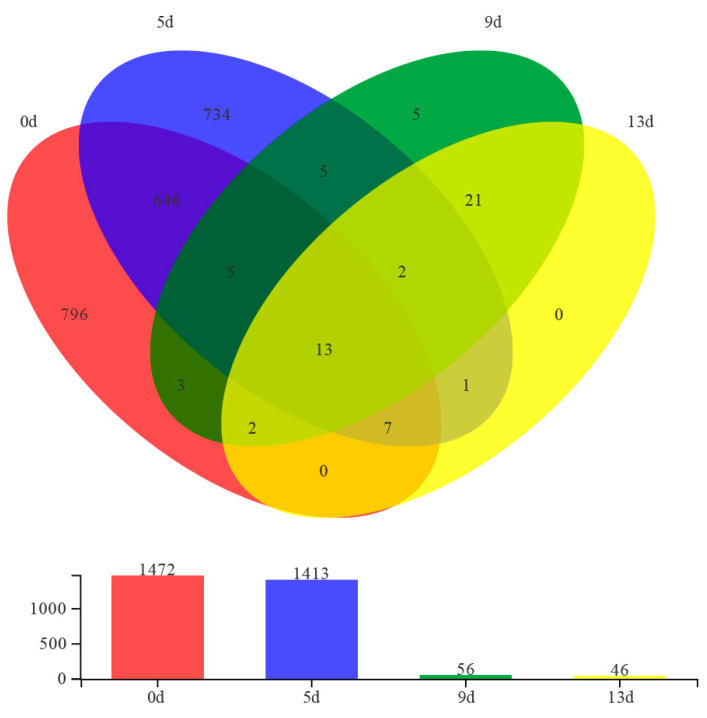
Venn diagram for bacteria OTUs in samples during storage (0d = red, 5d = blue, 9d = green, 13d = yellow).

**Figure 3 foods-12-02479-f003:**
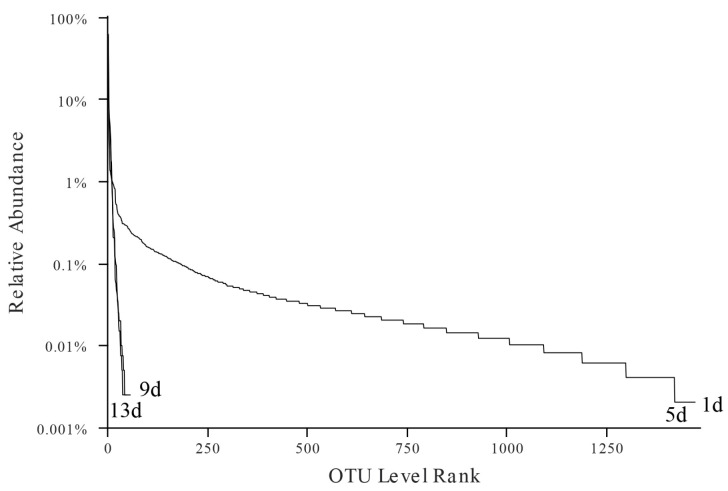
Rank-abundance curve showing bacterial community diversity during storage at 4 °C.

**Figure 4 foods-12-02479-f004:**
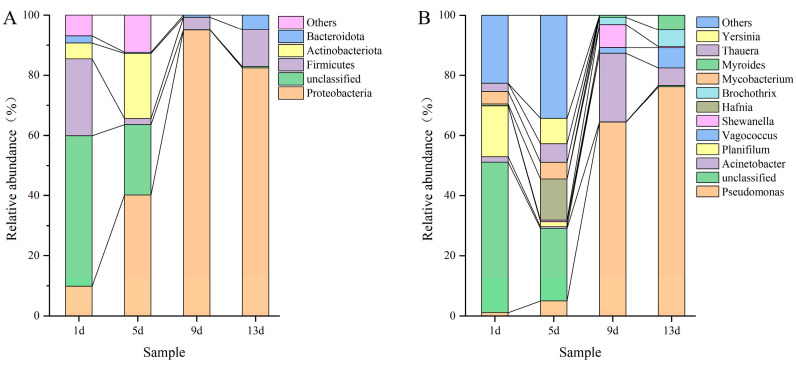
Percent of major bacterial community abundance at the phylum level (**A**) and the genus level (**B**).

**Figure 5 foods-12-02479-f005:**
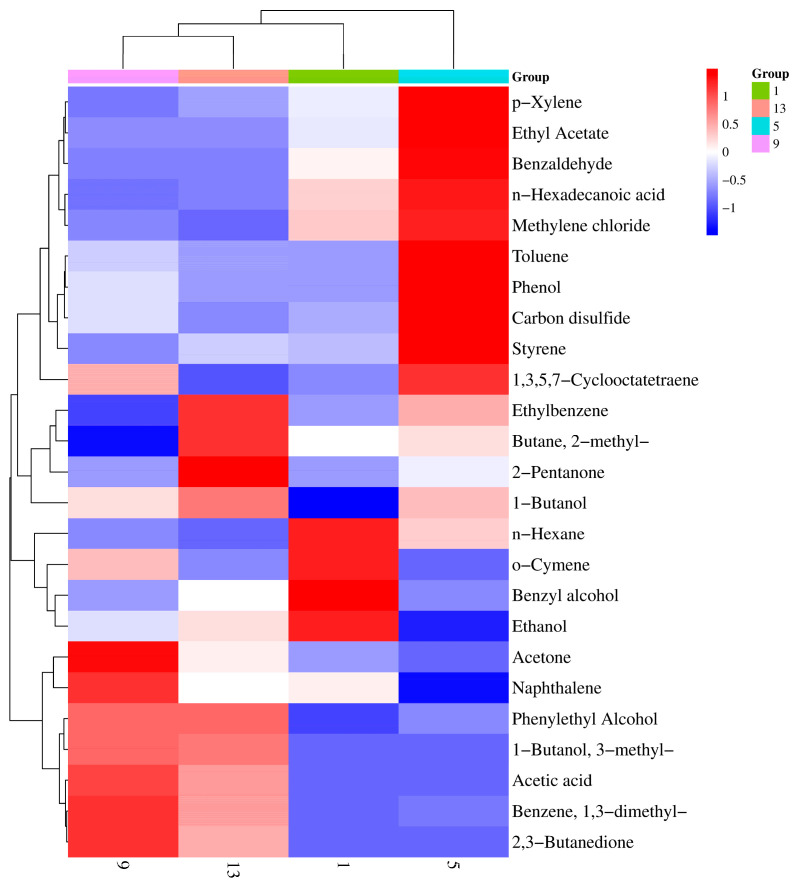
Heat map analysis was used to analyze the changes of volatile components of samples. Note: Blue represents negative correlation, red represents positive correlation, and the darker the color, the stronger the correlation.

**Figure 6 foods-12-02479-f006:**
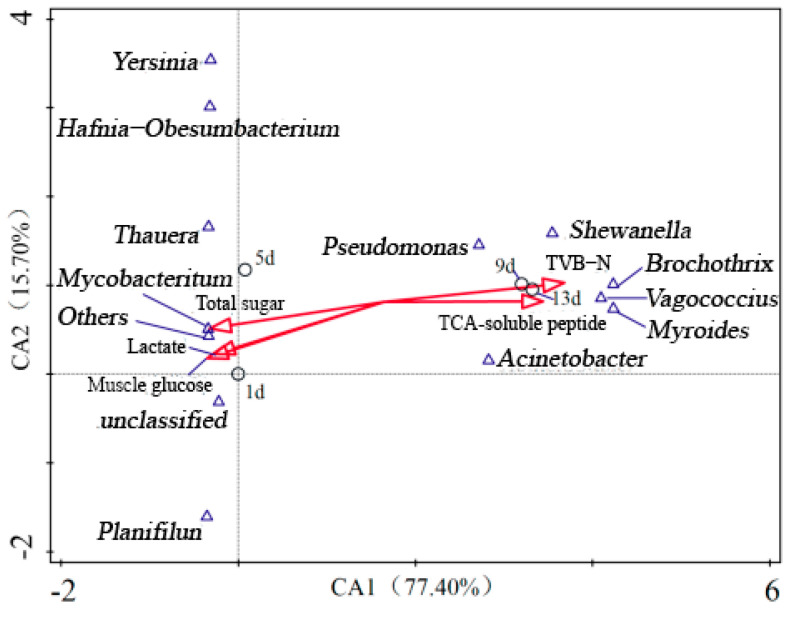
Correlation analysis of microbial species with protein and carbohydrates in samples.

**Figure 7 foods-12-02479-f007:**
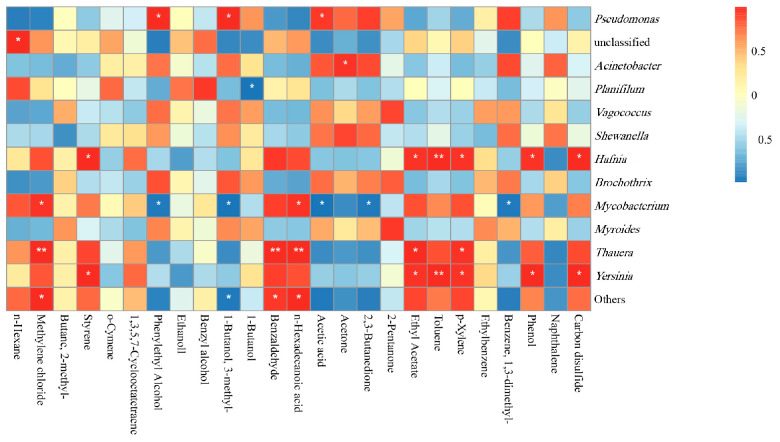
Correlation analysis of microbial species and VOCs in samples.

**Table 1 foods-12-02479-t001:** Changes in the value of color in samples during storage at 4 °C.

Color	Storage Time (d)							
	0	1	3	5	7	9	11	13
L*	43.96 ± 2.34 ^c^	48.03 ± 2.75 ^b^	48.17 ± 2.10 ^b^	48.33 ± 2.35 ^b^	49.21 ± 1.81 ^b^	49.28 ± 1.00 ^b^	52.47 ± 1.92 ^a^	54.35 ± 1.92 ^a^
a*	−2.13 ± 0.21 ^d^	−1.75 ± 0.11 ^c^	−1.92 ± 0.37 ^cd^	−1.64 ± 0.30 ^bc^	−1.31 ± 0.22 ^ab^	−1.32 ± 0.18 ^ab^	−1.20 ± 0.31 ^a^	−1.71 ± 0.40 ^c^
b*	0.27 ± 0.55 ^c^	0.95 ± 1.85 ^bc^	1.26 ± 0.60 ^abc^	1.99 ± 1.13 ^ab^	2.16 ± 1.00 ^ab^	1.56 ± 0.56 ^abc^	2.71 ± 1.64 ^a^	2.23 ± 1.17 ^ab^

Note: The same lowercase letter in the same column indicates that there is no difference in color between samples at different storage times (*p* > 0.05).

**Table 2 foods-12-02479-t002:** Change in BA contents in samples during storage at 4 °C.

BAs (mg/kg)				Storage Time (d)				
	0	1	3	5	7	9	11	13
TRP	0.08 ± 0.00 ^b^	0.29 ± 0.04 ^a^	ND	ND	ND	ND	ND	ND
PEA	0.04 ± 0.00 ^f^	0.09 ± 0.01 ^ef^	0.14 ± 0.02 ^e^	0.15 ± 0.00 ^e^	0.24 ± 0.02 ^d^	0.61 ± 0.01 ^c^	1.93 ± 0.08 ^b^	3.35 ± 0.04 ^a^
PUT	0.04 ± 0.01 ^h^	1.58 ± 0.04 ^g^	2.46 ± 0.13 ^f^	7.95 ± 0.19 ^e^	12.57 ± 0.38 ^d^	22.10 ± 0.26 ^c^	41.41 ± 0.20 ^b^	73.23 ± 0.36 ^a^
CAD	0.57 ± 0.19 ^h^	2.25 ± 0.18 ^g^	8.51 ± 0.09 ^f^	15.59 ± 0.23 ^e^	26.52 ± 0.44 ^d^	31.68 ± 0.24 ^c^	54.25 ± 0.30 ^b^	85.44 ± 0.25 ^a^
HIS	0.21 ± 0.01 ^f^	0.62 ± 0.04 ^e^	0.68 ± 0.03 ^de^	0.72 ± 0.01 ^cd^	0.76 ± 0.05 ^bc^	0.79 ± 0.03 ^ab^	0.81 ± 0.06 ^ab^	0.85 ± 0.04 ^a^
TYR	1.07 ± 0.06 ^h^	1.58 ± 0.06 ^g^	1.91 ± 0.07 ^f^	4.51 ± 0.08 ^e^	7.01 ± 0.09 ^d^	8.96 ± 0.05 ^c^	39.69 ± 0.07 ^b^	100.96 ± 0.05 ^a^
SPD	2.55 ± 0.06 ^g^	3.11 ± 0.09 ^f^	5.60 ± 0.02 ^d^	6.56 ± 0.02 ^c^	11.53 ± 0.11 ^a^	7.86 ± 0.04 ^b^	4.33 ± 0.23 ^e^	3.20 ± 0.16 ^f^
SPE	4.15 ± 0.12 ^f^	5.32 ± 0.26 ^e^	6.43 ± 0.35 ^d^	7.82 ± 0.67 ^c^	18.21 ± 0.17 ^a^	12.11 ± 0.09 ^b^	7.32 ± 0.26 ^c^	4.12 ± 0.10 ^f^

Note: TRP: Tryptamine; PEA: Phenethylamine; PUT: Putrescine; CAD: Cadaverine; HIS: Histamine; TYR: Tyramine; SPD: Spermidine; SPE: Spermine. The same lowercase letter in the same column indicates that there is no difference in color between samples at different storage times (*p* > 0.05).

**Table 3 foods-12-02479-t003:** Bacterial Alpha diversity index in samples during storage at 4 °C.

Sample	Reads	Shannon	Simpson	Ace	Chao1	Coverage
1d	52,332	5.62	0.02	1487.71	1482.78	99.89%
5d	52,940	6.02	0.01	1437.25	1440.78	99.86%
9d	44,778	1.62	0.36	89.05	82.25	99.96%
13d	44,771	1.55	0.40	57.57	61.00	99.98%

Note: Operational taxonomic units (OTUs) were the classified operational units.

## Data Availability

The authors declare that the dataset used and analyzed during the current study can be made available upon reasonable request.

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
