# Peer review of "Dynamic Changes of the Microbial Community and Volatile Organic Compounds of the Northern Pike (Esox lucius) during Storage"

_foods, 2023, doi:10.3390/foods12132479_

Round 1

Reviewer 1 Report

Line 23-24  Are the Vagococcus, Acinetobacter, and Mycoides associated with improper
practices?

Line 25-26  This sentence is vague. Please, rewrite.

I suggest a reorganization in the abstract, mainly in describing the result.

Methods

Section 2.6  Please, give a brief description of the procedure.

Lines 108 and 127  Insert symbol “×” before g force.

Line 170  Which statistical test was used?

Lines 170-171 - This sentence does not belong to this section.

Lines 171-172  Please, indicate the statistical test.

I recommend separating Figure 1.

Section 3.1 - Please, describe the results of the sensory test.

From which day were the fillets not acceptable? How can we correlate sensory tests with
microbiological analysis?

Figure 2  Describe the results shown in this figure.

Line 401  Please, explain why the ethyl acetate was higher than in other studies.

Lines 414- 417  Rewrite the sentence.

From your findings, how can we improve the shelf-life?

The English language is adequate. 

Reviewer 2 Report

The manuscript by Shang and co-authors describes the dynamics of changes of chemical compounds associated with the quality of food in relation to the changes in composition of the microbial community during storage of Northern pike fillets. The spoilage of food products, concerned with the activity of microorganisms is the serious economic problem, especially for fish products, therefore this work might have a significant input in food technology. The authors performed the analysis of sensory characteristics, biogenic amines and volatile organic compounds during the storage of pile fillets in the refrigerator during 0-13 days. In parallel, they performed the profiling of microbial communities inhabiting fillets and analyzed the dynamics of changes. In general, the manuscript makes a good impression, but contains a significant number of inaccuracies and deficiencies that need to be corrected.

Abstract and intro

L20: please, correct “high-through sequence analysis” to “high-throughput amplicon sequencing”

L46: “However, only some microorganisms are able to adapt to harsh environments, grow and reproduce.” Please, specify, which environments, described in the paper con be considered as “harsh”? Is it storage conditions (+4C)?

L49-54: “A small part of these microorganisms can secrete…” Almost all microorganisms secrete enzymes and secondary metabolites. Please, rephrase. Also please, make a more detailed description of the term “SSO”, currently it is not clear enough.

Materials and Methods

L90-93. Please, give more details regarding L*, a* and b* of CIELAB color space. In the presented form that might be not clear to the reader.

L158: Please, describe the DNA isolation procedure

L160-162: Please, describe the parameters V3V4 amplification in more detail (source of amplification reagents, thermocycling conditions, instrument etc.)

L164-167: Please, describe the library preparation method

L258-259: “A total of 194821 reads were generated, with an average of 97410 reads per sample…“ According to the numbers only two samples were sequenced. Please, check the read counts.

Figure5. Please increase the font size, or the whole figure. The names of the compounds are not readable. Also, please explain the numerical scale in the figure legend.

Figure 6. Please, describe the correlation analysis procedure.

3.7 Paragraph. It would be also interesting to see the correlations with other measurements described in the paper.

Conclusion

L417: The term “the main microbiota” is not correct. Might be changed to: “the most abundant microbial taxa were … at the phylum level. “

L425-426: Authors state, that: “Although H2S producing bacteria and Brochothrix thermophilus are considered to be important spoilage microorganisms, they are not suitable for shelf-life estimation.” However, I had not found in the text any proposal, how the specific bacteria counts might serve as the criterion for shelf-life estimation. Sensory assessment should be easier, shouldn’t it? Please, explain in details, or remove this statement from the manuscript.

The text is full of spelling and stylistic errors. Just a few examples:

L17-18: “Protein and sugar were used in large quantities by microorganisms in the later stage.”

“Used” should be changed to “detected”

L22-23: “A small number of microbiota composed of” sounds not really good. Might be changed to “The minor fraction of microbial consortium consisted of…”

L123: “The extraction method of biogenic amine in samples was based on Zhuang[22] and made appropriate adjustments.” Should be changed to “We used the extraction method of biogenic amine in samples described Zhuang and co-authors [22] with appropriate adjustments”

L367: “Protein degradated” instead of “Protein degraded”

It should be noted that this list is far from exhaustive, so the manuscript needs the extensive English editing.

Reviewer 3 Report

The authors present a match of different techniques for the control of fish shelf life. In general the content of the manucript id good but I would suggest some modification and changes in order to clarify some parts

It would be good to have a line of conclusions in the abstract

In the last part of the introduction in particular in the AIMs paragraph between L57 to L63 please indicate all the technique you have use to reach your goals with the indication of all avrebiation words that should be stated the first time it appears in the text, as for example  “tvb-n, tca-soluble peptide”

Please maintain the consistency of the abbreviation indications for the name of the techniques all should be written in capitals letters check the first line of the abstract.

L 66-68 please rephrase

L72 why the authors didn’t check all the parameters in all sampling times? Please explain

L75 please rephrase

L 75: “ Briefly, sensory accept ability test of samples was evaluated by a panel consisting of 6 trained laboratory members(three men and three women, 21 to 35 years old), according to a five-point scale.” What does it mean exactly? All men were 21 and all women 35 or it is the age range of all panel members? Please rephrase to add more clarity to the sentence

L92 “Detection method reference manual.” what does it means please clarify

L95 a space is needed between al and the first bracket

L95 “The determination of TCA-soluble peptides referred to the method of Zhuang et al[22], made some appropriate adjustments. “ please rephrase to make it clearer

L 96 a space is needed between 27 and ml

L118 what DNS  stand for? Reagent please correct

L 123 Zhuang[22] space needed before the bracket

L125 10.0mL 0.4M  sparationsg between numbers and units are needed please make an extensive check through all the manuscript 

L147 the extraction process was performed in isotherm or does the temperature of the sample changed during the extraction time? Please indicate

In figure 1 there are several orthographic errors  please check “mussel” may it be muscle? (Check it along the manuscript) Please change Storage times for Storage time and check the separation between the bracket and the precedent word

3.2. Color change  

Regarding the explanation have you consider the oxidation of the fatty content of the fish flesh since the values of b seems to mote to the yellow color and the values of a to the green part?

In general an extensive revision of the English grammar is needed paying particular attention to the verb tenses, some parts of the manuscript are quite difficult to understand and not very readable.
